# Ranking-Incentivized Document Manipulations for Multiple Queries

## ABSTRACT

In competitive retrieval settings, document publishers (authors) modify their documents in response to induced rankings so as to potentially improve their future rankings. Previous work has focused on analyzing ranking-incentivized document modifications for a *single query*. We present a novel theoretical and empirical study of document modification strategies applied for potentially improved ranking for *multiple queries*; e.g., those representing the same information need. Using game theoretic analysis, we show that in contrast to the single-query setting, an equilibrium does not necessarily exist; we provide full characterization of when it does for a basic family of ranking functions. We empirically study document modification strategies in the multiple-queries setting by organizing ranking competitions. In contrast to previous ranking competitions devised for the single-query setting, we also used a neural ranker and allowed in some competitions the use of generative AI tools to modify documents. We found that publishers tend to mimic content from documents highly ranked in the past, as in the single-query setting, although this was a somewhat less emphasized trend when generative AI tools were allowed. We also found that it was much more difficult with neural rankers to promote a document to the highest rank simultaneously for multiple queries than it was with a feature-based learning-to-rank method. In addition, we demonstrate the merits of using information induced from multiple queries to predict which document might be the highest ranked in the next ranking for a given query.

## 1 INTRODUCTION

In the Web search setting, publishers (authors) of documents are sometimes incentivized to have their documents highly ranked by search engines for some queries. The basic motivation is the fact that documents at top ranks often attract most user engagement [19]. As a result of their ranking incentives, publishers might modify their documents in response to rankings induced by a search engine so as to improve their future rankings. The search setting then becomes competitive [21] with corpus dynamics driven in part by ranking-incentivized document modifications. These modifications are often referred to as search engine optimization (SEO) [14].

Lately, there has been a growing body of work on studying ranking-incentivized document modifications, and more generally, on competitive retrieval [3, 7, 8, 11–13, 23, 25, 32, 34, 36, 37, 39,

40]. This renewed interest in the often dubbed *adversarial retrieval* setting [6] can be attributed in part to a large volume of work on adversarial machine learning in the era of neural network models [16–18, 20, 30, 35, 42, 44].

The focus of some recent work on competitive retrieval is on algorithmic attacks on ranking functions; e.g., substituting terms with their synonyms [7, 8, 13, 23, 25, 34, 37, 39]. There is also an emerging thread of work on analyzing, and improving, the robustness of ranking functions to ranking-incentivized document modifications [3, 7, 11, 36, 40]. Another type of work, which is our focus in this paper, is analyzing human strategies of ranking-incentivized document manipulations [32] and their resultant effect on the corpus [3, 13, 32]. For example, it was shown theoretically and empirically that a prominent (worthwhile) strategy of publishers is to mimic content in documents that were highly ranked in the past for queries of interest [32], which results in an *herding* effect [13].

Almost all recent work on competitive retrieval has focused on a single-query setting; i.e., assuming a publisher modifies her document to improve its ranking for a single query[1]. However, publishers often opt to have their document highly ranked for multiple queries; e.g., those representing a topic in the document. Accordingly, in this paper we present the first — to the best of our knowledge — theoretical and empirical study of ranking-incentivized document-modification strategies of (human) publishers opting to promote their documents for multiple queries.

Using game theoretic analysis we show that when publishers modify their documents to improve their rankings for *multiple* queries, there is not necessarily an equilibrium. This result implies instability: a potentially endless document modification race which can have negative effects on users; e.g., documents at top ranks consistently change not due to pure editorial considerations but rather due to ranking incentives. This theoretical result is in contrast to recent findings about the existence of an equilibrium when documents are modified for a *single* query [32].

We then fully characterize the conditions for an equilibrium in the multiple-queries setting for a basic family of ranking functions. Furthermore, we show that *best response dynamics* [27] does not necessarily converge to an equilibrium in case it exists. Best response dynamics is a process where each publisher modifies her document to attain the best possible ranking given the documents written by all other publishers. The implication is that reaching a steady state in case it is achievable might call for an external intervention; e.g., of the search engine.

We next present an empirical study of document manipulation strategies applied by publishers that opt for improved ranking for multiple queries representing the same information need (topic). The study is based on ranking competitions we organized between students. Our competitions were inspired by those organized for

---

[1]We discuss in Section 2 the two exceptions: algorithmic attacks for topically-related queries [25] and the corpus effect of modifying documents for two queries (topics) [3].

the single-query setting [11, 13, 32]. The competitions we report differ from previous competitions [11, 13, 32] not only by the virtue of having publishers modify their documents for multiple queries rather than a single one, but also by two additional aspects. First, our competitions are the first to employ not only a feature-based learning-to-rank (LTR) method [24] as the ranking function, but also a neural ranker [22]. Second, in some of our competitions, we encouraged the participants to use generative AI tools (e.g., GPT [29]) to help modify their documents. Indeed, the SEO industry is using generative AI tools (https://www.seoclarity.net/research/impact-generative-ai), and our (controlled) study sheds light on usage strategies and implications. The competitions were approved by ethics committees. The data of the competitions and our code will be made public upon publication of this paper.

Analysis of the ranking competitions revealed that, as is the case for the single-query setting [32], publishers tend to mimic content from documents that were previously highly ranked. This trend was more prominent when publishers were not allowed to use generative AI tools. We also found that the neural ranker's rankings for different queries representing the same information need were more diverse than those of the LTR method. Accordingly, it was much harder for publishers to have their documents highly ranked for multiple queries with the neural ranker.

Finally, as in work on the single-query setting [32], we pursue the task of predicting which document among those not ranked first in the last ranking will become the most highly ranked assuming that the top ranked document is going to change. As it turns out, utilizing information induced from rankings for other queries representing the same information need can help improve prediction effectiveness for the query at hand. This is yet another difference between the multiple-queries setting and the single-query setting.

Our contributions can be summarized as follows:

- A theoretical and empirical study of document manipulations intended to improve ranking for *multiple* queries.
- A full game theoretic characterization of when equilibrium of document manipulation strategies exists.
- Showing that best response dynamics might not converge even when an equilibrium exists.
- Organizing novel ranking competitions where publishers compete for multiple queries. Additional novel aspects are using a neural ranker and allowing publishers to use generative AI tools.
- A method for predicting which documents among those not ranked first might be ranked first in the next ranking; the method uses information induced from other queries.

## 2 RELATED WORK

The work most related to ours is based on a game theoretic and empirical analysis of ranking incentivized document manipulations applied for a *single* query [32]. In contrast to our *multiple* queries settings, an equilibrium in the single-query setting always exists. Raifer et al. [32] and Goren et al. [13] found using ranking competitions that publishers tend to mimic content from documents highly ranked in the past for a query: a strategy justified by Raifer et al. [32] using a game theoretic analysis. We found a similar pattern in ranking competitions organized for the multiple-queries setting.

We use both a neural ranker and a feature-based ranker while Raifer et al. [32] used only the latter. In contrast to our work, Raifer et al. [32] did not allow the use of generative AI tools for document manipulation in their ranking competitions.

Several algorithmic attacks on neural retrieval methods have been recently reported [7, 8, 13, 23, 25, 34, 37, 39]. The attacks are almost always for a single query, except for an attack for topically related queries [25]. In contrast, in our game theoretic analysis, the document manipulations are changes of the relative emphasis of a document on different queries. This type of strategic manipulation has not been studied in previous work to the best of our knowledge. Furthermore, we are not aware of any other studies, as ours, of human document-modification strategies for multiple queries with or without generative AI tools.

The suboptimality of the probability ranking principle [33] in competitive retrieval settings was demonstrated using a game theoretic approach [3]. Publishers can write either a single topic document or a two-topics document with equal emphasis for the topics. Since the resulting ranking game has a finite number of players (publishers) and a finite number of actions a player can play (i.e., documents they can write), by Nash's theorem [26], a mixed-strategies equilibrium exists. That is, documents are stochastically created by applying a distribution over a document set. The setting is less realistic than the one we address here where publishers can write a document with different emphasis (represented as a real number) on different queries. Our resulting games have an infinite number of actions a publisher can take, and we show that there is not necessarily an equilibrium.

## 3 GAME THEORETIC ANALYSIS

To analyze the search setting when publishers modify their documents to have them highly ranked for multiple queries, we use game theory. We consider publishers as players in a ranking game: the documents they publish are their strategies and the search engine's ranking function is the mediator [21].

As noted above, some previous work used game theory to analyze the single-query search setting where publishers opt to promote their documents for a single query [32]; single-peaked ranking functions were used [32]. Several sparse retrieval methods are single peaked; e.g., cosine between tf.idf vectors and negative KL divergence over multinomial distributions [43]. Furthermore, considering the common technique of stuffing query terms in the document to improve its ranking [14], the retrieval function could also be viewed as single peaked with respect to query term occurrence. That is, moderately increasing query term occurrence usually results in monotonic increase of retrieval scores. However, as from a certain point, further increasing query term occurrence can result in increased penalty to retrieval scores when document quality measures are used [5]. There is also some empirical evidence in prior work [32] and ours (Section 5) that publishers might view the undisclosed ranking function as single peaked. When considering incomplete information about the ranker (unknown peak), the ranking game has a minimax regret equilibrium where publishers mimic content of documents that were previously the highest ranked [32].

Turning to our multiple-queries setting, we assume that a document has split emphasis on various queries; e.g., different passages

match different queries to varying degrees. We assume as Raifer et al. [32] a single peaked ranking function, and focus as Basat et al. [2] on ranking games where the retrieval function is disclosed (i.e., its peak is known). We show that the resulting situation in the multiple-queries setting with a disclosed ranking function is considerably more subtle than that in the single query setting where it is trivial to show that an equilibrium exists. Specifically, we prove that there is not necessarily an equilibrium in the multiple-query setting and fully characterize the cases when it exists.

## 3.1 Game Definition

Let $G = \langle n, m, p \rangle$ be a ranking game defined as follows: $n$ publishers (players) write and modify documents in a corpus $D$; a query set $Q$ contains $m$ queries: $\{q_1, \ldots, q_m\}$. Each player writes a single document $d \in D \overset{def}{=} \{(d^1, \ldots, d^m) \in [0,1]^m : \sum_{j=1}^m d^j \leq 1\}$, where $d^j$ ($\in [0,1]$) is the relative emphasis of the document $d$ on information relevant to query $q_j$.[2]

Let $f : [0,1] \rightarrow [0,1]$ be a single peak function: there exists a unique $p \in [0,1]$ such that $f$ is monotonically increasing in $[0,p]$ and monotonically decreasing in $[p,1]$. The retrieval function $r : D \times Q \rightarrow [0,1]$ assigns document $d$ ($\in D$) the retrieval score $r(d, q_j) \overset{def}{=} f(d^j)$ with respect to query $q_j$ ($j \in \{1, \ldots, m\}$); documents are ranked in descending order of scores with ties broken arbitrarily.

We will refer to document $d_i$ written by player $i$ as the (pure) **strategy** of player $i$ in the game[3]. That is, the strategy $d_i$ is defined by the choice of relative emphasis on each query in the document $d_i$. A **strategy profile** $s \overset{def}{=} (d_1, \ldots, d_n)$ is the tuple of pure strategies of the players in the game; i.e., the set of documents in the corpus.

The **utility** of a player $i$ who wrote document $d_i$, denoted $U_i(s)$, is the number of queries for which $d_i$ is ranked first; $d_i$'s rank for a query depends on the strategies of all other players (i.e., their documents). To summarize, $G$ is a strategic game where $n$ players (publishers) opt to maximize their documents' ranks for $m$ queries.

A strategy profile $s$ is a (pure) **Nash equilibrium** if there is no incentive for any player $i$ to change her strategy given the strategies $\{d_j\}_{j \neq i}$ of all other players; i.e., a change will result in reduced utility.

## 3.2 Game Analysis

Our goal is to find a full characterization of the existence (or lack thereof) of pure Nash equilibria for $G$, given $n$, $m$ and $p$.

It is easy to see that for a small enough $p$, there exists a pure Nash equilibrium where the documents published by all players have the same emphasis on all queries:[4]

**LEMMA 3.1.** If $p \leq \frac{1}{m}$, then the profile where all players write $d = (p, \ldots, p)$ is a pure Nash equilibrium.

To extend our analysis for any $p$ ($\in [0,1]$), we start by analyzing the case of two players ($n = 2$). This case has an interesting property,

because in any pure Nash equilibrium, the two players have the same utility. They can attain this utility, for example, by using the same strategy (i.e., writing the same document):

**LEMMA 3.2.** Let $n = 2$ and $m > n$. If $s = (d_1, d_2)$ is a pure Nash equilibrium, then $U_1(s) = U_2(s) = \frac{m}{2}$.

**PROOF.** Observe that for any profile $s = (d_1, d_2)$, $U_1(s) + U_2(s) = m$. Assume by contradiction that $U_1(s) \neq U_2(s)$, w.l.o.g. $U_1(s) < U_2(s)$; i.e., $U_1(s) < \frac{m}{2} < U_2(s)$. If player 1 adopts player 2's strategy, i.e., writes $d_1' = d_2$ then $U_1(d_1', d_2) = U_2(d_1', d_2) = \frac{m}{2} > U_1(d_1, d_2)$; hence, $s = (d_1, d_2)$ is not a pure Nash equilibrium — a contradiction. □

We use Lemma 3.2 to fully characterize in Theorem 3.3 the existence of pure Nash equilibrium for $G$ when $n = 2$ (i.e., two players).

**THEOREM 3.3.** If $n = 2$ and $m > n$, then the game $G$ has a pure Nash equilibrium iff $p \leq \frac{1}{m-1}$.

**PROOF.** We provide here a proof sketch, with the comprehensive proof provided in Appendix B. Say a strategy profile $(d_1, d_2)$ is a pure Nash equilibrium; let $d_2 = (d_2^1, \ldots d_2^m)$, and assume w.l.o.g. that $d_2^1 \geq d_2^2 \geq \ldots \geq d_2^m$. We show that there cannot be two queries $q_i$ and $q_j$ such that $d_2^i < p$ and $d_2^j < p$. This allows to show that if there is an equilibrium, then $p \leq \frac{1}{m-1}$. If $p \leq \frac{1}{m}$, then by Lemma 3.1 the profile where both players publish $d = (p, \ldots, p)$ is a pure Nash equilibrium. We show that for $\frac{1}{m} < p \leq \frac{1}{m-1}$ the profile where $d_1 = d_2 = (p, \ldots, p, 1 - p \cdot (m-1))$ is a pure Nash equilibrium. □

To extend the analysis to a game with more than two players, we start by considering the case of $m > n$:

**THEOREM 3.4.** The game $G = \langle n, m, p \rangle$ with $n < m$ has a pure Nash equilibrium iff $p \leq \frac{1}{\lceil \frac{2 \cdot m}{n} - 1 \rceil}$.

**PROOF.** The full proof is mostly technical and fully detailed in Appendix B. The central idea is to show that in any pure Nash equilibrium the following properties must hold: (i) the highest ranked document $d$ for each query $q_j$ has $d^j \geq p$; and, (ii) for each player there is at most one query for which her document is ranked first solely. For any $p > \frac{1}{\lceil \frac{2 \cdot m}{n} - 1 \rceil}$, these properties cannot hold, and therefore there is no pure Nash equilibrium. Conversely, we show that if $p \leq \frac{1}{\lceil \frac{2 \cdot m}{n} - 1 \rceil}$ then we can construct a pure Nash equilibrium as follows.

Given some strategy profile $s = (d_1, \ldots, d_n)$, let $h_j(s)$ denote the number of documents assigned the same highest retrieval score for query $q_j$ when $s$ is played. The strategy profile $s_{\text{alg}}$ obtained by the following algorithm is a pure Nash equilibrium :

---

**Algorithm 1** Equilibrium construction for $G = \langle n, m, p \rangle$ with $n < m$

---

Initialize $d_i^j = 0$ for all $i, j$.
$k = \left\lfloor \frac{1}{p} \right\rfloor$
**for** $t = 1$ to $k$ **do**
    **for** $i = 1$ to $n$ **do**
        $j^* = \arg\min_{j \in \{1, \ldots, m\}} h_j(d_1, \ldots, d_n)$
        $d_i^{j^*} = p$
    **end for**
**end for**

---

□

---

[2]In reference to work on focused retrieval [10], $d^j$ can be defined as the portion of the text in $d$ that is marked as relevant to query $q_j$.

[3]There is also a notion of mixed strategies where players apply a distribution over pure strategies. Herein, we focus on pure strategies.

[4]Recall that the retrieval function has a single peak at $p$.

In the case $n \geq m$, it is easy to see that a pure Nash equilibrium exists for every $p$. Specifically, w.l.o.g. each (disjoint) group of $\lfloor \frac{n}{m} \rfloor$ players is assigned to an arbitrarily different query $q_j$ among $q_1, \ldots, q_m$; each player in this group writes a document $d$ with $d^j = p$ and $d^k = 0$ for $k \neq j$. Together with Theorem 3.4, we arrive to the final result:

COROLLARY 3.5. $G=\langle n, m, p \rangle$ has a pure Nash equilibrium iff $p \leq \frac{1}{\max\{\lceil \frac{2 \cdot m}{n} - 1 \rceil, 1\}}$.

### 3.3 Best response dynamics

The **best response** of a player to the strategies played by all other players (i.e., their documents) is the strategy (document) that maximizes her utility. Iteratively finding best responses until convergence (if exists) — a process known as **best response dynamics** — is a standard mechanism for reaching Nash equilibria, specifically in *potential games* [27]. In these games, the best response dynamics is guaranteed to converge to an equilibrium. We now show that in our ranking games the best response dynamics does not necessarily converge. Hence, these are not potential games. An important implication is that in order to achieve stability (specifically, equilibrium), the game might need external intervention.

Consider the game $G = \langle n, m, p \rangle$ for $n = 2, m = 3$ and $p = \frac{1}{2}$. We show that the best response dynamics does not converge in this game, although by Theorem 3.3 an equilibrium exists. Let $s_t$ denote the strategy profile played at round $t$ by two players. The following sequence of strategies is a best response dynamics given an initial strategy profile $s_0 = ((0.3, 0.4, 0), (0.2, 0.3, 0.5))$: $s_1 = ((0.3, 0.4, 0), (0.4, 0.5, 0.1))$, $s_2 = ((0.5, 0.3, 0.2), (0.4, 0.5, 0.1))$, $s_3 = ((0.5, 0.3, 0.2), (0, 0.4, 0.3))$. For example, since player 1's document at round 0 is $(0.3, 0.4, 0)$ then player 2's document at round 1, $(0.4, 0.5, 0.1)$, attains the maximal possible utility (3) as it outranks player 1's document for all three queries. Player 1's best possible utility given player 2's document from round 1, $(0.4, 0.5, 0.1)$, is 2, which is obtained by writing, for example, a document $(0.5, 0.3, 0.2)$ at round 2, and so forth. Since $s_0$ and $s_3$ are symmetric, the dynamics does not converge.

## 4 RANKING COMPETITIONS

Our next goal is to empirically analyze ranking competitions in which publishers (document authors) modify documents with respect to multiple queries ($m > 1$). To the best of our knowledge, there are no datasets that support this type of analysis. There are two publicly available datasets [12, 32] that resulted from ranking competitions. However, in these competitions, documents were modified with respect to a single query. Thus, we created a new dataset by organizing our own ranking competitions inspired by those organized in [12, 32]. International and institutional ethics committees approved the competitions. The participants signed a consent form and could withdraw at any time.

In addition to having publishers compete for multiple queries rather than a single one, we introduced two novel aspects in our competitions. The first is the ranking function. While in previous competitions only a feature-based learning-to-rank (**LTR**) method was used [12, 32], we also used a neural (**NEU**) ranker (BERT-based [28]). The ranking functions are described below. The second aspect is the use of AI tools to modify documents in some of the

**Table 1: Overview of the four ranking competitions.**

| Competition | Ranking Function | AI Tools |
|---|---|---|
| LTR∧AI | LambdaMART | ✓ |
| LTR∧¬AI | LambdaMART | ✗ |
| NEU∧AI | BERT | ✓ |
| NEU∧¬AI | BERT | ✗ |

competitions. In total, we organized four ranking competitions which differ by the ranking function applied and/or whether generative AI tools were allowed: **LTR∧AI**, **LTR∧¬AI**, **NEU∧AI**, and **NEU∧¬AI**. Table 1 provides an overview of the competitions.

Each competition lasted for ten rounds; in each round, 30 games were held, each with respect to a different backstory (topic) from the UQV dataset [1].[5] Each topic was represented using $m = 3$ queries: the query selected to be the focus of the backstory and two additional query variants selected from all those provided for the backstory. The same 30 topics were used in all four competitions. Refer to Appendix A.1 for further details about the topic and query selection procedures.

**Competition structure**. Before the start of the game, the participants were provided with three queries representing the topic of the game, the topic description (backstory) and an example of a relevant document. This document was generated using GPT-3.5, as described in Appendix A.1. The participants were then asked to write and submit their own document. In each round of the game, the participants were presented with the documents submitted in the previous round and the rankings induced over these documents for each of the three queries. They were encouraged to further modify their document to improve its ranking in the next round. All the documents were in plain text format, limited to 150 words. Refer to Appendix A.2 for detailed competition guidelines.

A group of 84 undergraduate and graduate students enrolled in an information retrieval course participated in our competitions. Each student participated in three (repeated) games and submitted one document per game in a round. We made sure that the three games were selected from at least two different competitions and that there was no overlap between the topics of these games. Each game included exactly five participants: a combination of students and bots. Two or three students participated in every game together with two bots that used GPT-3.5 to create documents, as explained in Appendix A.3. If a game had only four participants, i.e., two students and two bots, we added a third static bot that published the initial example document in every round. To prevent any potential bias, the students' identities were anonymized, and they were not informed about the use of bots. Unless otherwise mentioned, our analysis is performed for the students' documents only.

To encourage students to participate and modify their documents so as to attain high rankings in the competitions, we offered bonus points for their course grades. We note that the students could have earned a perfect grade in the course without participating in the competitions. The bonus per game was awarded based on the median reciprocal rank of a student's document across the three queries in the game. For example, if a student's document was ranked first, second, and third across the queries, they would earn

---

[5]The dataset includes query variants for TREC's 2013-2014 Web track topics (201-300).

half a point for that game. The points were accumulated over the student's three games per round and over the ten rounds.

Our dataset includes a total of 2520 documents submitted by students (847 of which are unique), 2400 documents generated by bots (1833 of which are unique), and 30 documents that served as the initial document per topic. We will make the dataset publicly available upon the publication of this work.

**Ranking functions**. As noted above, we experimented with two ranking functions in our competitions: LTR and NEU. For our learning-to-rank (LTR) approach, as in previous work [12, 32, 36], we use LambdaMART [41] via the RankLib toolkit[6]. Unless otherwise specified, the implementation details of this model, including the training procedure and setting of hyper-parameter values, are the same as those of Vasilisky et al. [36]. The model was trained on the Combined dataset [36], which includes data from the two previous ranking competitions [12, 32]. Each document was represented using a nine-dimensional feature vector. Some of the features are estimates of the similarity between the query and document, and the rest are query-independent document quality measures shown to be effective for Web retrieval [4]. The features are: (i) **TF**: sum of query term frequencies in the document, (ii) **NormTF**: TF normalized by the document's length, (iii) **BM25**: BM25 similarity between the query and document, (iv) **LMIR**: language-model-based similarity between the query and document[7], (v) **BERT**: BERT-based similarity between the query and document[8], (vi) **LEN**: document's length, (vii) **FracStop**: percentage of terms in the document that are stopwords[9], (viii) **StopCover**: percentage of stopwords on a stopword list that appear in the document, and (ix) **ENT**: entropy of the document's term distribution. For our neural ranking function (NEU), we used BERT as in feature (v) above. Thus, BERT served both as a feature in LTR and as a standalone ranking function. We note that the ranking functions were not disclosed to the students, who also did not know that four ranking competitions were held.

**Relevance and quality judgements**. Each student document was judged by three crowd workers in CloudResearch's connect platform [15] for binary relevance to the topic and by five workers for content quality, with 0.75 and 0.49 inter-annotator agreement rates (free-marginal multi-rater Kappa), respectively. All workers were native English speakers. More than 97% of all student documents were judged relevant by at least two out of three annotators. This high percentage is in accordance with previous reports on competitions for a single query [13, 32], albeit a bit elevated. Although students were not instructed to write relevant documents, the documents were short and students modified them for multiple queries representing the same topic. We note that our focus is on ranking-incentivized manipulations and not on ranking effectiveness.

Five annotators judged each student document for content quality using the categories: valid, keyword stuffed and spam (useless). When using AI tools, 89% and 71% of the documents were judged to be valid by at least three and at least four out of the five annotators, respectively. When not using AI tools, the percentages were 85% and 61%, respectively. This attests to the merit of using AI tools

for rank promotion while maintaining content quality. Only 9% and 6% of the documents created with no AI tools and with AI tools, respectively, were judged as keyword stuffed by at least three annotators. Almost none of the documents were judged as spam. The distribution of quality judgments was not different between the LTR and NEU competitions.

# 5 EMPIRICAL ANALYSIS

## 5.1 Analysis of Strategies

**Feature Values**. Inspired by work on the single-query setting [32], we examine the document modification strategies in our multiple-queries setting: we analyze the changes in the documents' feature values along competition rounds. Recall that three document rankings were induced per topic, each for one of the three queries representing the topic. In this analysis, we examine each query (document ranking) separately. We focus on changes in documents that lost (i.e., were not ranked first) for at least four consecutive rounds before reaching the first rank (**L**). We compare the average feature values of the winners (i.e., the highest ranked documents) per query ($\mathbf{W_q}$) with the average feature values of two types of losing documents (*L*): those whose feature value was lower than or equal to that of the winning document four rounds prior to their win ($\mathbf{L \leq W_q}$), and those whose feature value was higher ($\mathbf{L > W_q}$). We also present for reference the average feature values of the (at most three) winning documents per topic ($\mathbf{W_t}$). Figure 1 presents the results for representative query-independent and query-dependent features of LTR. Figures for other features exhibit the same patterns and are omitted as they convey no additional insight.

We see in Figure 1 that regardless of the ranking function (LTR or NEU), the use of generative AI tools (AI or ¬AI) and the initial feature values ($L \leq W_q$ or $L > W_q$), with a few exceptions, the features of the losing documents tend to gradually converge to those of the winners. This finding is in line with results for the setting where publishers modified their documents for a single query [32]. Contrary to Raifer et al. [32], we see that the feature values of the winning documents changed throughout the competition rounds. This might be attributed to the fact that documents in our competitions were modified for three different queries. For example, say a document was ranked first for some query. The student who wrote the document may have continued to modify it to improve its ranking for the other two queries, and thus affected its feature values. We delve deeper into this point in Section 5.2.

We also note that changes of winners' feature values conceptually echo our game theoretic findings in Section 3: in the single-query setting an equilibrium is guaranteed in contrast to the multiple-queries setting. Hence, the single-query setting is more likely to reach a stable state than the multiple-queries setting[10]

We observe in Figure 1a a general upward trend for the query-independent features: LEN, StopCover, and ENT; the trends are more pronounced for LTR∧AI and LTR∧¬AI. Hence, the documents became longer (LEN) with increased content diversity (ENT) and stopwords occurrence (StopCover). Interestingly, Raifer et al. [32]

---

[6]https://sourceforge.net/p/lemur/wiki/RankLib/
[7]Unigram language models with Dirichlet smoothing ($\mu = 1000$) are used.
[8]We used BERT-Large fine-tuned for passage ranking on MS MARCO [28].
[9]The NLTK stopword list was used in our experiments: www.nltk.org/nltk_data/.

---

[10]We hasten to point out that the game theoretic analysis was performed for single peak functions. The rankers used in the competition are not single peaked.

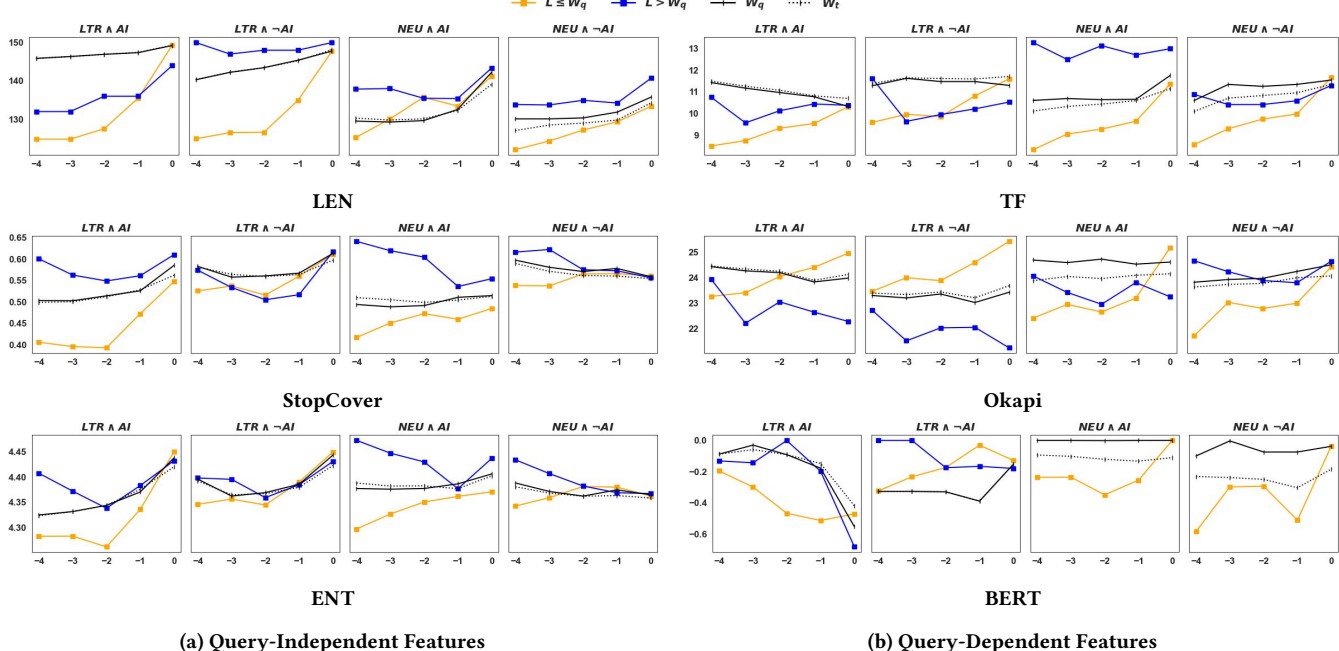

(a) Query-Independent Features

(b) Query-Dependent Features

**Figure 1: Averaged feature values of documents that lost in at least four consecutive rounds before winning. The documents are grouped based on whether their feature values four rounds before the win were lower than or equal to ($L \leq W_q$) or higher than ($L > W_q$) the values of the winners for the given query. $W_q$ and $W_t$: the averaged feature values of the corresponding winner per query and all the winners per the same topic, respectively. x-axis: the (negative) number of rounds leading up to a win. Note: in NEU∧AI and NEU∧¬AI, when BERT is used as a standalone ranking function, the feature values of the losing documents are always lower than those of the winning document; therefore, $L > W_q$ is not shown.**

found a downward trend for ENT when students modified documents for a single query. We therefore hypothesize that when competing for a single query there is an attempt to focus the content for this query, while competing for multiple queries results in increased diversity so as to better cover the query set.

We see in Figure 1b that the query-dependent feature values for $L \leq W_q$ (where the initial feature value of the losing document was lower than or equal to that of the winner) increased along the rounds, whereas the values for $L > W_q$ (where the initial feature value of the losing document was higher than that of the winner) somewhat fluctuated and even decreased at times. This implies that in the former case, the students gradually increased the number of query term occurrences in their documents, but in the latter case, the students were careful not to add too many query terms, possibly to avoid keyword stuffing which they were warned about. Furthermore, this finding potentially suggests that from the point of view of students, the ranking function might be single peaked with respect to query term occurrence and the features based on it.

Finally, we can see that the differences between $W_q$ and $W_t$ are smaller for LTR than for NEU. This finding might be attributed to the fact that for LTR, a winning document was more likely to win for multiple queries, resulting in fewer unique winning documents per topic. We present further support to this claim in Section 5.2.

**Prompts**. To gain additional insights about the document modification strategies, we asked the students to share the prompts they

used for the generative AI tools. We collected 44 prompts from rounds 8 to 10 of the competitions. The prompts typically included instructions to create a document that would be ranked high for the specified queries. All prompts, except for three, had no mention of the reward mechanism used in the competitions. As an example, one of the three prompts included the text: "Since the score is given by the median ranking among 3 queries, we can focus on maximizing the rankings on two queries."

Out of the 44 prompts, 29 included example documents: the initial example document (2 prompts), the student's document from the previous round (15 prompts), at least one of the winning documents from the previous round (24 prompts), or even all the documents in the previous round (5 prompts). This further indicates that the students considered winning documents a valuable source of information about the undisclosed ranking functions.

## 5.2 Ranking Functions

Thus far, we studied the document modification strategies in our competitions. We now shift our focus to examining the effect of the ranking functions used. We first analyze the differences between rankings induced by a function for the three queries representing a topic. We computed for each ranking function the RBO similarity (with $p = 0.9$) [38] between the document rankings induced for each pair of queries per topic. We then averaged these values over the pairs per topic and over topics. Figure 2 presents the results for

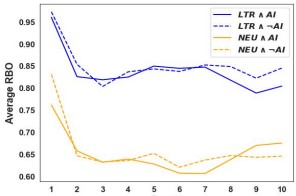

**Figure 2: Average RBO similarity between all pairs of rankings induced per topic, averaged over topics per round.**

**Table 2: Docs: percentage of winning documents that won for exactly $x$ queries per topic. Rank: for documents that won for $x$ queries per topic, the average ranks with respect to the $3 - x$ queries for which they did not win.**

|  | LTR∧AI | | LTR∧¬AI | | NEU∧AI | | NEU∧¬AI | |
|---|---|---|---|---|---|---|---|---|
|  | Docs | Rank | Docs | Rank | Docs | Rank | Docs | Rank |
| $x = 1$ | 17.9% | 2.2 | 17.7% | 2.4 | 62.9% | 3.4 | 65.7% | 3.5 |
| $x = 2$ | 17.0% | 2.1 | 15.2% | 2.3 | 28.5% | 3.3 | 27.6% | 3.2 |
| $x = 3$ | 65.1% | - | 67.0% | - | 8.6% | - | 6.7% | - |

each round of the four competitions. We see that the similarities for NEU∧AI and NEU∧¬AI are considerably and consistently lower than those of LTR∧AI and LTR∧¬AI, suggesting that NEU (BERT) is much more sensitive to the queries used than LTR (LambdaMART). Previous research demonstrated BERT's sensitivity to document modifications in competitions held for a single query [36].

We further study the effect of the ranking functions on the competitions in Table 2 by examining documents that won for at least one of the queries per topic. We present the percentage of these documents that won for exactly one, two or three queries. Additionally, we present the average ranks of these documents in rankings induced for queries for which these documents did not win. We can see that for LTR, at least 65% of the winners won all three queries in a game, while for NEU, less than 9% did so. Furthermore, the average rank of documents for queries they did not win was, on average, higher when NEU was used than when LTR was used.[11] These findings demonstrate that the ranking function used affects the ability of a document to win for multiple queries per topic. That is, a document that won for one of the queries is more likely to win for additional queries for the same topic in competitions that employed LTR compared to those that employed NEU.

### 5.3 Generative AI Tools

We allowed the use of generative AI tools in some of our competitions. We next compare the documents created by students who were allowed to use such tools (AI) with those who were not allowed (¬AI). We use the cosine of tf.idf vectors to compute per topic the similarity between each document submitted by a student in a specific round and (i) documents in the previous round that won for at least one of the queries and (ii) all the other documents from the previous round. We averaged the similarities over the documents and grouped them by whether or not the students were

---

[11]The rank of the highest-ranked document is 1. Increased rank means lower ranking.

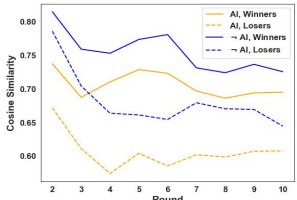

**Figure 3: Average similarity between *all* documents in a round *per topic* and (i) Winners: documents that won at least one query for the topic in the previous round and (ii) Losers: all other documents in the previous round.**

allowed to use AI tools. We see in Figure 3 that there were noticeable differences in the similarity values of AI and ¬AI. Specifically, the documents created without the help of AI tools in LTR∧¬AI and NEU∧¬AI were much more similar to documents submitted in the previous round, whether they were winners or not, than those created with the help of such tools in LTR∧AI and NEU∧AI. This finding indicates that using AI tools may lead to the creation of more diverse content. We also observe higher similarities with the winning documents than with the losing ones for both AI and ¬AI, further supporting our earlier findings that the documents, whether they were directly modified by students or generated using AI tools, tended to converge to the winners.

## 6 PREDICTING WINNERS

In this section, we address the challenge of predicting which document among all those that did not rank first in round $l - 1$ will be ranked the highest (win) in round $l$ if the winner indeed changes. The documents in our competitions were modified with respect to $m = 3$ queries in the query set $Q$ for a topic; yet, the prediction is performed separately for each query $q \in Q$. A similar challenge was previously tackled for competitions held for a single query [32].

For prediction, we represent each document as a 27-dimensional feature vector. The features we use can be divided into seven sets. The first two sets, **Micro** (12 features) and **Macro** (3 features), were proposed by Raifer et al. [32]. These features model properties of the document in round $l$ ($D_l$), the document published by the same author in round $l - 1$ ($D_{l-1}$), and the winning document in round $l - 1$ for the query $q$ ($W_{l-1}^q$). The features in Micro quantify the changes in $D_l$ with respect to $W_{l-1}^q$ by counting term additions and deletions. These features are defined for three groups of unique terms: query terms, stopwords, and terms that are neither query terms nor stopwords. For each group of terms, four features were defined: the number of terms in $W_{l-1}^q$ that were either (i) added to or (ii) deleted from $D_{l-1}$, and the number of terms that were not in $W_{l-1}^q$ and were either (iii) deleted from or (iv) added to $D_{l-1}$. The features in Macro are the similarities between $D_l$, $D_{l-1}$, and $W_{l-1}^q$; the similarity between a pair of documents was computed using the cosine of their tf.idf vector representations.

The features just discussed only consider the winner for the given query ($W_{l-1}^q$), disregarding the rankings induced for the other two queries and their corresponding winners. Accordingly, we introduce

an additional set of features, **TopicMacro**, that consider the group of all winners for the three queries in $Q$ ($W_{l-1}^Q$). We compute the similarities between $D_l$ and $W_{l-1}^Q$ and between $D_{l-1}$ and $W_{l-1}^Q$. We use cosine to compute the similarities and represent the documents in $W_{l-1}^Q$ via the centroid of their tf.idf vectors.

In the following two feature sets, we quantify properties of $D_{l-1}$'s ranks in the three rankings induced for all three queries per topic. The first set, **TopicRank**, includes three features: the minimum, median, and maximum rank of $D_{l-1}$ across the three queries in $Q$. The second set, **QueryRank**, includes three binary features that are set to 1 if $D_{l-1}$'s minimum, median, and maximum rank for the three queries is equal to its rank for the given query $q$; otherwise, they are set to 0.

The next feature set, **PrevChange**, includes three features that quantify past changes (improvements) in the document's ranks. The underlying assumption is that if a publisher improved her document's rank in round $l-1$, she might do so again in round $l$. To this end, we also examine the publisher's document in round $l-2$ ($D_{l-2}$). For each of the three queries, we compute the difference between the ranks of $D_{l-1}$ and $D_{l-2}$, and scale it by the maximal possible rank change given $D_{l-2}$'s rank in round $l-2$. We use the minimum, median, and maximum values across the three queries.

Finally, we found in Section 5.1 that the documents tended to become longer throughout the four competition rounds leading to a win; accordingly, the feature in the final set, **Len**, is $D_l$'s length.

## 6.1 Setup

Our dataset includes documents submitted by both students and bots. We discarded queries for which a bot generated the winning document. In addition, we discarded queries for which the winners in round $l-1$ ($W_{l-1}^q$) and round $l$ ($W_l^q$) were generated by the same publisher. This was done because we aimed to predict which of the losing documents would win in the following round. As a result, the number of queries per competition round ranged from 3 to 39.

We experimented with four different classifiers with the features defined above via the scikit-learn library [31]: logistic regression (**LReg**), linear SVM (**LSVM**), polynomial SVM (**PSVM**) and random forests (**RForest**). We applied min-max normalization to feature values per query. The document assigned the highest prediction score per query by a classifier was deemed the winner; the remaining documents were considered losers.

The features in PrevChange can only be computed when data for at least two previous rounds is available; therefore, our experiments were conducted for rounds 3 to 10. We used leave-one-out cross-validation over rounds to train our models and select parameter values. We used the queries from one of the rounds for testing and those from the remaining seven for training and validation. We repeatedly trained a model on data from six of the seven rounds with different parameter configurations and validated the effectiveness over the seventh round. We selected the parameter values that yielded the highest prediction effectiveness on average across the seven validation rounds. Then, we trained a final model using data from all seven rounds and applied it to the held-out test round (fold). We repeated this entire procedure for each test fold. The models were trained separately for each of the four competitions.

**Table 3: Prediction effectiveness. All differences between our classifiers (LReg, LSVM, PSVM, RForest) and the baselines are statistically significant. Bold: best result in a row.**

| | | Rand | QMaj | TMaj | AllW | AllL | Raifer | LReg | LSVM | PSVM | RForest |
|---|---|---|---|---|---|---|---|---|---|---|---|
| LTR∧AI | Acc | 0.654 | 0.435 | 0.503 | 0.466 | 0.534 | 0.696 | 0.822 | 0.853 | **0.895** | 0.843 |
| LTR∧AI | F1 | 0.622 | 0.397 | 0.497 | 0.635 | 0.000 | 0.678 | 0.810 | 0.843 | **0.885** | 0.832 |
| LTR∧¬AI | Acc | 0.707 | 0.398 | 0.504 | 0.480 | 0.520 | 0.772 | **0.886** | 0.870 | 0.846 | 0.837 |
| LTR∧¬AI | F1 | 0.697 | 0.368 | 0.497 | 0.648 | 0.000 | 0.763 | **0.879** | 0.863 | 0.839 | 0.831 |
| NEU∧AI | Acc | 0.669 | 0.532 | 0.468 | 0.464 | 0.536 | 0.766 | 0.815 | 0.798 | 0.806 | **0.855** |
| NEU∧AI | F1 | 0.643 | 0.495 | 0.588 | 0.634 | 0.000 | 0.747 | 0.800 | 0.782 | 0.790 | **0.842** |
| NEU∧¬AI | Acc | 0.556 | 0.481 | 0.500 | 0.491 | 0.509 | 0.722 | 0.750 | 0.750 | **0.796** | 0.750 |
| NEU∧¬AI | F1 | 0.547 | 0.471 | 0.592 | 0.658 | 0.000 | 0.717 | 0.742 | 0.742 | **0.790** | 0.743 |

We use **Acc** (percentage of correctly classified winners and losers) and **F1** (harmonic mean of precision and recall), averaged over queries and test folds, to measure prediction effectiveness; the former served to select parameter values. Statistically significant prediction differences were determined across queries using the two-tailed paired t-tests with $p \leq 0.05$. LReg, LSVM and PSVM were trained with L1 regularization; the regularization parameter was selected from {1, 10, 50, 100}. The degree of the polynomial kernel in PSVM was selected from {2, 3, 4, 5}. For RForest, the number of trees and leaves were set to values in {10, 50, 100, 500} and {10, 20, 30}, respectively. All other parameters were set to default values [31].

## 6.2 Results

We compare in Table 3 the prediction effectiveness of our four classifiers (LReg, LSVM, PSVM, and RForest) with that of five baselines[12]: (i) Random (**Rand**) : the winner is randomly selected, (ii) QueryMajority (**QMaj**) : the winner is the document whose publisher had the highest number of past wins for the given query; ties were broken randomly, (iii) TopicMajority (**TMaj**): the winner is the document whose publisher had the highest number of past wins for all three queries of the given topic; ties were broken randomly, (iv) AllWinners (**AllW**): all the documents are predicted to be winners (only one document is correctly classified per query), (v) AllLosers (**AllL**): all the documents are predicted to be losers (all but one of the documents are correctly classified per query). We also report the effectiveness of the classifiers when using only the two feature sets proposed by Raifer et al. [32] (Micro and Macro). For this baseline, we show the results only for the classifier (LReg, LSVM, PSVM, or RForest) with the highest Acc per competition.

The documents in our competitions were modified with respect to three queries. Table 3 shows that when using only the two feature sets proposed by Raifer et al. [32], which use information only about the query at hand, the performance surpasses that of all baselines. This attests to the effectiveness of these feature sets, whether the documents are modified with respect to a single query as in [32] or with respect to multiple queries as in our setting. Furthermore, we see that when adding features that quantify various properties of all three queries and the corresponding rankings, all our classifiers consistently and statistically significantly outperform all the baselines, including that of Raifer et al. [32]. This attests to the merits of using information about other queries for the same topic to predict for a given query.

---

[12]Except for TopicMajority, all the baselines were also used by Raifer et al. [32].

To further examine the effectiveness of the different feature sets, we performed ablation tests, removing one of the sets each time. We anzlye the results for PSVM, for which the best overall performance was attained for two of the four competitions in Table 3. Actual numbers are omitted as they convey no additional insight and due to space considerations. Except for Macro, removing each feature set results in at least one statistically significant drop in performance, attesting to the complementary nature of our features. Comparing Macro and TopicMacro, the drop for the latter is more substantial. These findings suggest that these two feature sets might be somewhat correlated and that TopicMacro, which considers the winners for all queries of the topic in the previous round, is somewhat more informative in our setting.

## 7 CONCLUSIONS

We presented a novel theoretical and empirical study of the competitive retrieval setting where document authors modify documents to improve their ranking for multiple queries. We showed using game theory that in contrast to past work on document modifications for a single query, in the multiple-queries setting there is not necessarily an equilibrium and characterized the cases when it exists. We organized novel ranking competitions where publishers modified documents for multiple queries representing the same topic. We used both neural and non-neural rankers, and allowed the use of generative AI tools in some competitions. We found that publishers use content from documents highly ranked in the past, but to a somewhat reduced extent when they apply AI tools. Finally, we addressed the task of predicting which document will be promoted to the highest rank in the next ranking round. We demonstrated the merits of using information induced from other queries representing the same topic to this end.

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

## A RANKING COMPETITIONS

### A.1 Topics, Queries, and Initial Documents

We selected from the UQV dataset [1] topics with commercial intent that were more likely to incite competition[13]. For each topic, three queries were used. The first query was the focus of the backstory, while the other two were selected from all the available query variants as follows. To ensure that the variants were of high quality, we retrieved 1000 documents per variant from the ClueWeb12 category B corpus using LMIR via the Indri toolkit[14]. We filtered out variants with average precision (AP) lower than 0.05. In addition, to ensure that the queries were diverse enough, we computed the Jacquard similarity between the terms of each pair of queries and the RBO similarity (with $p = 0.9$) [38] between the corresponding retrieved document lists. These two scores were combined using Reciprocal Rank Fusion (RRF) with $k = 60$ [9]. Finally, from the remaining query variants, we iteratively selected the following query with the lowest average similarity to the already selected queries.

We provided the students with an example of a relevant document for each topic. We generated these documents using GPT-3.5 with a prompt per topic that included the backstory and the three selected queries. We continued generating the document until it was deemed relevant by four annotators. We used the same document per topic across all four competitions, resulting in 30 documents.

### A.2 Additional Guidelines

The students were instructed to create high quality plain text documents without using any formatting tags, hyperlinks, or special characters. They were advised to avoid using slang or informal language and were warned against keyword stuffing. We mentioned that using such techniques would result in a penalty, although no actual penalty was imposed. Copying parts of other students' documents or Web pages was allowed as long as the document also included original content. Documents that were copied completely from other documents received no reward. We noticed after the third round that some students were submitting the initial example document without modifications. Thus, we prohibited submitting the initial documents starting from the fourth round. Students who still used these documents did not receive points. We emphasized that the goal was to promote documents in rankings regardless of their relevance to the information needs expressed by the queries.

---

[13]The selected topics are 201, 203, 204, 209, 210, 211, 212, 218, 226, 228, 233, 235, 244, 245, 246, 249, 250, 252, 255, 258, 261, 262, 268, 272, 274, 281, 283, 289, 291, and 296.
[14]www.lemurproject.org/indri

### A.3 Bots

We used bots in our competitions to increase the number of players and to make the games more dynamic. The bots generated documents using GPT-3.5. The prompts asked to create documents no longer than 150 terms, but some of the generated documents were longer. In these cases, we removed sentences from the end of a document until its length was between 140 and 150 terms. If no sentence could be removed without making the document shorter than 140 terms, we truncated the last sentence to 150 terms.

We provided our bots with the backstory, the three queries representing the backstory, and the content and ranks of the bot's document in the previous round. In addition, we provided information about other submitted documents that varied across bots based on two parameters. The first parameter determined which of the documents submitted in a round were accessible to the bot. We tested bots that had access to all submitted documents (including their ranks), the document with the highest median rank across the three queries (excluding the actual ranks), and the bot's document only. The second parameter determined the number of rounds for which the above information was made available: last round or last three rounds. We used six bot configurations in our competitions, each corresponding to combinations of the two parameters. Each bot was assigned to ten different topics and competed in all ten rounds of the game.

The documents generated by bots were generally ranked lower than those written by students. The average rank for the bots' documents was 3.2, while the students' documents had an average rank of 2.6. Moreover, the bots' documents won less: 67.9% of the winning documents were written by students, compared to only 23.5% generated by bots.[15]

## B PROOFS

### B.1 Equilibrium in Two-Player Games

THEOREM 3.3. *If $n = 2$ and $m > n$, then the game $G$ has a pure Nash equilibrium iff $p \leq \frac{1}{m-1}$.*

PROOF. Assume there exists some strategy profile $(d_1, d_2)$ that is a pure Nash equilibrium. Let $d_2 = (d_2^1, \ldots d_2^m)$, and assume w.l.o.g. that $d_2^1 \geq d_2^2 \geq \ldots \geq d_2^m$.

Suppose there are two queries $q_i$ and $q_j$ such that $d_2^i < p$ and $d_2^j < p$. Define $d_t \in D$ such that $d_t^i = \min\{d_2^i + \frac{d_2^2}{2}, p\}$, $d_t^j = \min\{d_2^j + \frac{d_2^1}{2}, p\}$, $d_t^1 = 0$ and $d_t^k = \leq \sum_{l=1}^m d_2^l \leq 1$.

If player 1 writes $d_t$, then $d_t$ will be ranked (i) first (solely) for $q_i$ and $q_j$, (ii) below the first rank for $q_1$, (iii) the same as $d_2$ for the rest of the queries. Thus, if player 1 writes $d_t$ her utility is $\frac{m-3}{2} + 2 = \frac{m+1}{2}$. By Lemma 3.2, if $(d_1, d_2)$ is a pure equilibrium then the utility of player 1 is $\frac{m}{2}$. Therefore, the move from writing $d_1$ to writing $d_t$ is beneficial for player 1, which is a contradiction to the fact that $(d_1, d_2)$ is a Nash equilibrium. Thus, we conclude that if $(d_1, d_2)$ is a pure Nash equilibrium then there is at most one query $q_j$ such that $d_2^j < p$. Now, if $\frac{1}{m-1} < p$, then there exists no such document $d_2 \in D$. Therefore, $p \leq \frac{1}{m-1}$. Hence, we proved that if there is a Nash equilibrium, $p \leq \frac{1}{m-1}$. We now turn to show that for any $p \leq \frac{1}{m-1}$ there is a pure Nash equilibrium.

---

[15]The percentages do not add up to 100 as we excluded from this analysis the static bots that submitted the initial document in all competition rounds.

If $p \leq \frac{1}{m}$, then by Lemma 3.1 the profile where both players publish $d = (p, \ldots, p)$ is a pure Nash equilibrium. If $\frac{1}{m} < p \leq \frac{1}{m-1}$, the profile where $d_1 = d_2 = (p, \ldots, p, 1 - p \cdot (m - 1))$ is a pure Nash equilibrium as we show next. (Note that $(m-1) \cdot p \leq 1$ and therefore the strategies ($d_1$ and $d_2$) are well defined.)

W.l.o.g. assume that player 1 has an incentive (in terms of increased utility) to move from writing $d_1$ to writing some $d_t$. For every query $q_j$ s.t. $j < m$, if $d_t^j \neq p$, then $d_t$ cannot be ranked first because the retrieval function reaches its single peak at $p$. Thus, the only way to increase utility would be to have $d_t$ ranked first for query $q_m$. This can only be achieved by increasing the emphasis on $q_m$ but as a result decreasing the emphasis on some other query $q_j$, where $j < m$. Consequently, $d_t$ will be ranked (i) first (solely) for query $q_m$, (ii) ranked second for query $q_j$ (iii) the same as $d_2$ for the rest of the queries. Therefore, player's 1 utility will be $\frac{m-2}{2} + 1 = \frac{m}{2}$, which is the same as her utility when writing $d_1$; i.e., there was no incentive to move to $d_t$. □

## B.2 Game Analysis

*Notations.* Let $w_j(s)$ represent the value of the document ranked highest in query $q_j \in Q$ under the strategy profile $s$. We use $s_{-i}$ to denote the strategies of all players except player $i$, and $w_j(s_{-i})$ to denote the highest value of the documents in query $q_j$ when player $i$ doesn't participate. Hence, if player $i$ is not ranked highest in query $q_j$, or if the highest rank is shared among multiple players, then $w_j(s_{-i}) = w_j(s)$. However, if player $i$ uniquely tops the ranking in query $q_j$, then $w_j(s_{-i}) < w_j(s)$.

Player $i$ is called a *winner* in query $q_j$ if $d_i^j = w_j(s)$. Define $J_i(s)$ as the set of queries in which player $i$ is a winner under strategy profile $s$:

$$J_i(s) := \{q_j \in Q : d_i^j = w_j(s)\}$$

We begin by identifying a set of strategies in the game that are weakly dominated. This initial step enables us to focus solely on strategies in subsequent proofs where $d_i^j \leq p$ for every player $i$ and query $q_j \in Q$.

**Lemma B.1.** *For every player $i$, publishing a document $d$ with $d_j > p$ for some query $q_j \in Q$ is weakly dominated.*

**Proof.** Let $d_i$ be a strategy of player $i$, where $\max_{q_j \in Q} d_i^j > p$. Consider the strategy $d_t$ which is defined by $d_t^j = \min\{d_i^j, p\}$. In queries where $d_i^j > p$, the document $d_t$ will be ranked first by $r$. There will be no change in the other queries. Therefore, $U_i(s_{-i}, d_i) \leq U_i(s_{-i}, d_t)$ and $d_i$ is weakly-dominated by $d_t$. □

To facilitate the analysis of the game dynamics, we introduce some key definitions. The *residual* of a player represents their maximum potential unused resource when deviating to another strategy profile that allows them to maintain victory in the same set of queries. Formally,

*Definition B.2 (Residual).* The *residual* of player $i$ in the game $G = \langle n, m, p \rangle$ under the strategy profile $s$ is given by:

$$y_i(s) := 1 - \sum_{q_j \in J_i(s)} w_j(s_{-i})$$

The residual can be equivalently calculated by:

**Lemma B.3.** *Let $G = \langle n, m, p \rangle$ and $s$ be some pure strategy profile.*

$$1 - y_i(s) = \min_{d' \in D_i} \left\{ \sum_{q_j \in Q} d'_j \,\middle|\, J_i(s) = J_i(s_{-i}, d') \right\}$$

This expression computes the minimal total sum of document values ($\sum_{q_j \in Q} d^j$) needed for player $i$ to continue winning the same set of queries when deviating from their current strategy. Although the set of winning queries remains unchanged, the utility for player $i$ might vary because the number of winners in each query could increase.

The following lemma proposes a necessary condition for the existence of a pure equilibrium by considering the residual of a player relative to the minimal value of the winner among all queries where they are currently not winning. A larger residual indicates an opportunity for a profitable deviation, as it reflects the presence of unused resources that could be strategically allocated to win at least one additional query.

**Lemma B.4.** *Let $G = \langle n, m, p \rangle$ and $s$ be some pure strategy profile. For every player $i$, if*

$$y_i(s) > \min\{\min_{q_j \notin J_i(s)} \{w_j(s)\}, p\}$$

*then $i$ has a profitable deviation, indicating that $s$ is not an equilibrium.*

**Proof.** Consider a player $i_1$ for which

$$y_{i_1}(s) > \min_{q_j \notin J_{i_1}(s)} \{w_j(s_{-i_1})\} \tag{1}$$

We utilize the equivalent definition of the residual from Lemma B.3. Define $d$ as:

$$d = \arg \min_{d' \in D_{i_1}} \left\{ \sum_{q_j \in Q} d'_j \,\middle|\, J_{i_1}(s) = J_i(s_{-i_1}, d') \right\}$$

Let $q_{j_t} = \arg \min_{q_j \notin J_{i_1}(s)} \{w_j(s_{-i_1})\}$.

Consider a deviation to $d_t$ defined by:

$$d_t^j = \begin{cases} \min\{w_j(s_{-i_1}) + \epsilon_0, p\}, & q_j \in J_{i_1}(s) \cup \{q_{j_t}\} \\ 0, & \text{otherwise} \end{cases} \tag{2}$$

where $\epsilon_0 = \frac{y_{i_1}(s) - w_{j_t}(s_{-i_1})}{|J_{i_1}(s)| + 1}$.

To validate the deviation, we demonstrate that $\sum_{q_j \in Q} d_t^j \leq 1$:

$$\sum_{q_j \in Q} d_t^j \leq \sum_{q_j \in J_{i_1}(s) \cup \{q_{j_t}\}} w_j(s_{-i_1}) + \epsilon_0$$
$$= y_{i_1}(s) + \sum_{q_j \in J_{i_1}(s)} w_j(s_{-i_1}) \tag{3}$$
$$= y_{i_1}(s) + 1 - y_{i_1}(s)$$
$$= 1$$

According to the definition of $d_t$, for every query $q_j \in J_{i_1}(s)$, it follows that $q_j$ also belongs to $J_{i_1}(s_{-i_1}, d_t)$. After the deviation, the number of winners ($h_j(s)$) in queries in $J_{i_1}(s)$ can only decrease, leading to an increase in player $i_1$'s reward for them. Furthermore, player $i_1$ is now ranked first in query $q_{j_t}$. As a result, player $i_1$ experiences a minimum increase in overall utility by at least $\frac{1}{n}$, making the deviation profitable. □

Consider a scenario in a game where a player is one of multiple winners (i.e., $h_j(s) \geq 2$) for a particular query $q_j$. If the winning score $w_j(s)$ is not at the peak value $p$, a minor strategic adjustment could enable him to outscore other players, thus becoming the sole winner. Such a shift would increase his utility since he would no longer share the reward for the query. Consequently, at equilibrium, this player must have strategically allocated all available resources to maximize his outcome across contested queries. This implies that his residual resources should be zero.

LEMMA B.5. *Let $G = \langle n, m, p \rangle$ and $s$ be some pure strategy profile. Define $I_t = \{i : \exists q_j \in J_i(s) \text{ s.t. } w_j(s) < p, h_j(s) \geq 2\}$. Then for every player $i \in I_t$,*
$$y_i(s) = 0$$

PROOF. Assume by contradiction that there exists a player $i_1 \in I_t$ with a residual $y_i(s) > 0$, indicating that they have unutilized resources. Consider the following strategic deviation:

$$d_t^j = \begin{cases} \min\{w_j(s_{-i}) + \frac{y_i(s)}{|J_i(s)|}, p\}, & q_j \in J_{i_1}(s) \\ 0, & q_j \notin J_{i_1}(s) \end{cases}$$

This deviation is feasible, since:

$$\sum_{j=1}^{m} d_t^j = \sum_{j \in J_i(s)} \min\{w_j(s_{-i}) + \frac{y_i(s)}{|J_i(s)|}, p\}$$
$$\leq \sum_{j \in J_i(s)} w_j(s_{-i}) + y_i(s)$$
$$\leq 1$$

By definition of $I_t$, there exists a query $q_{j_1} \in J_{i_1}(s)$ for which $w_{j_1}(s) < p$ and $h_{j_1}(s) \geq 2$. Player $i_1$'s deviation would increase his score sufficiently to potentially become the unique winner in query $q_{j_1}$, thereby increasing their utility by at least $\frac{1}{2}$. This gain in utility contradicts the assumption of equilibrium, thus proving that $y_i(s) = 0$ for all $i \in I_t$. □

The definition of the residual allows us to measure the quantity of available resources that a player has when playing each strategy. We introduce another strategic metric, $x_i(s)$, which quantifies the resources that a player allocates to queries they do not currently win. Formally,

$$x_i(s) := \sum_{q_j \notin J_i(s)} d_i^j$$

This metric measures the extent to which a player invests in preventing other winners from significantly reducing the winning value, thus establishing a floor on the resources required for others to secure a win in those queries. Although $x_i(s)$ does not directly impact player $i$'s own utility, it plays a critical role in limiting the potential for other players to deviate. By strategically allocating his residual resources, a player can influence the residual values of other players, thereby affecting their ability to deviate profitably from their current strategies.

We now derive an upper bound for $x_i(s)$, stating that the amount of resources a player uses in queries where he is the loser is no more than his overall available resources.

LEMMA B.6. *Let $G = \langle n, m, p \rangle$ and $s$ be a pure equilibrium profile. For every player $i$:*
$$y_i(s) \geq x_i(s)$$

PROOF. By definition of $x_i(s)$,

$$1 \geq \sum_{j=1}^{m} d_i^j$$
$$= \sum_{q_j \in J_i(s)} d_i^j + \sum_{q_j \notin J_i(s)} d_i^j$$
$$= \sum_{q_j \in J_i(s)} w_j(s) + x_i(s)$$
$$\geq \sum_{q_j \in J_i(s)} w_j(s_{-i}) + x_i(s)$$

Rearranging, we get that the residual of player $i$ satisfies:

$$y_i(s) = 1 - \sum_{q_j \in J_i(s)} w_j(s_{-i}) \geq x_i(s)$$

□

The proof theorem 3.4 relies on two key lemmas. The first states that in an equilibrium, the winner's value must be at least $p$. The second lemma states that every player is the unique winner in no more than one of his winning queries. In order to prove the first lemma, we begin by lower bounding the number of queries in which a player wins in equilibrium.

LEMMA B.7. *Let $G = \langle n, m, p \rangle$ and $s$ be a pure equilibrium profile. Let $k \in \mathbb{N}$ s.t. $\frac{1}{k+1} < p \leq \frac{1}{k}$. Then for every player $i$,*
$$|J_i(s)| \geq k$$

PROOF. Assume by contradiction there exists a player $i_1$ s.t. $|J_i(s)| \leq k - 1$. Let $q_{j_1}$ be a query in which $i_1$ is not ranked first, i.e. $q_{j_1} \notin J_i(s)$. Consider the following deviation strategy:

$$d_t^j = \begin{cases} \min\{d_i^j, p\}, & q_j \in J_{i_1}(s) \\ p, & q_j = q_{j_1} \\ 0, & \text{otherwise} \end{cases}$$

Given that $p \leq \frac{1}{k}$, the deviation is feasible:

$$\sum_{q_j \in Q} d_t^j = p + \sum_{q_j \in J_{i_1}(s)\}} \min\{d_i^j, p\}$$
$$\leq p + |J_{i_1}(s)| \cdot p$$
$$\leq p + (k - 1) \cdot p$$
$$\leq 1$$

Player $i_1$ maintains their winning status in all queries $q_j \in J_{i_1}(s)$ under $d_t^j$, and now potentially wins $q_{j_1}$ as well. Moreover, $h_j(s) \geq h_j(s_{-i_1}, d_t)$ for every $q_j \in J_{i_1}(s)$. Therefore, $i_1$'s utility after deviation is:

$$U_{i_1}(s_{-i}, d_t) = \sum_{q_j \in J_i(s_{-i}, d_t)} \frac{1}{h_j(s_{-i_1}, d_t)}$$

$$= \sum_{q_j \in J_i(s)} \frac{1}{h_j(s_{-i_1}, d_t)} + \frac{1}{h_{j_1}(s_{-i_1}, d_t)}$$

$$\geq \sum_{q_j \in J_i(s)} \frac{1}{h_j(s)} + \frac{1}{h_{j_1}(s_{-i_1}, d_t)}$$

$$= U_{i_1}(s) + \frac{1}{h_{j_1}(s_{-i_1}, d_t)}$$

$$> U_{i_1}(s)$$

indicating a profitable deviation of player $i_1$, by contradiction to $s$ being an equilibrium. □

Next, we leverage Lemma B.7 to derive an upper bound on $x_i(s)$ for all players. W.l.o.g., we assume for the rest of the section that $w_1(s) \geq \ldots \geq w_m(s)$.

LEMMA B.8. *Let $G = \langle n, m, p \rangle$ and $s$ be a pure equilibrium profile. For every player $i$,*

$$x_i(s) \leq 1 - \left\lfloor \frac{1}{p} \right\rfloor \cdot \min_{q_j \in Q} w_j(s)$$

PROOF. By the definition of $x_i(s)$, for every player $i$:

$$x_i(s) = \sum_{q_j \notin J_i(s)} d_i^j$$

$$= \sum_{q_j \notin J_i(s)} d_i^j + \sum_{q_j \in J_i(s)} d_i^j - \sum_{q_j \in J_i(s)} d_i^j$$

$$= \sum_{q_j \in Q} d_i^j - \sum_{q_j \in J_i(s)} d_i^j$$

$$\leq 1 - \sum_{q_j \in J_i(s)} d_i^j \qquad (4)$$

$$= 1 - \sum_{q_j \in J_i(s)} w_j(s)$$

$$\leq 1 - \sum_{q_j \in J_i(s)} w_m(s)$$

$$= 1 - |J_i(s)| \cdot w_m(s)$$

Applying Lemma B.7, we know $|J_i(s)| \geq \left\lfloor \frac{1}{p} \right\rfloor$. Thus, plugging into Equation 4,

$$x_i(s) \leq 1 - |J_i(s)| \cdot w_m(s) \leq 1 - \left\lfloor \frac{1}{p} \right\rfloor \cdot w_m(s)$$

□

## B.3 Equilibrium in Game with $m > n$

The proof of Theorem 3.4 relies on the following Lemma, which asserts that the winning value in every query across the entire game must meet or exceed the peak value $p$.

LEMMA B.9. *Let $G = \langle n, m, p \rangle$ such that $n < m$ and $s$ be a pure equilibrium profile. Then for every query $q_j \in Q$,*

$$w_j(s) \geq p$$

We begin by proving the Lemma for the following special scenario. Consider the case where there exist a query where the winning value is lower than the peak, and there is only one player winning in that query. We show that the winner in that query must be the unique winner in all the queries where he wins. Then, we use the upper bounds we derived on $x_i(s)$ to show there must exist a player with profitable deviation. The proof hinges on the interplay between the number of queries a player wins and their values in queries they do not win.

LEMMA B.10. *Let $G = \langle n, m, p \rangle$ such that $n < m$, $p \leq \frac{1}{2}$, and let $s$ be a pure equilibrium profile. Then for every query $q'_j$ where $h'_j(s) = 1$:*

$$w_{j'}(s) \geq p$$

PROOF. Assume for contradiction there exists a query $q_{j_1}$ where $h_{j_1}(s) = 1$ and $w_{j_1}(s) < p$. Let $i_1$ be the sole winner of $q_{j_1}$.

By the definition of $y_i(s)$, since $i_1$ is the only winner in query $q_{j_1}$:

$$y_{i_1}(s) = 1 - \sum_{q_j \in J_{i_1}(s)} w_j(s_{-i_1})$$

$$> 1 - \sum_{q_j \in J_{i_1}(s)} w_j(s)$$

$$\geq 1 - \sum_{q_j \in Q} d_{i_1}^j \qquad (5)$$

$$\geq 0$$

Next, assume by contradiction that there exists another query $q_{j_2} \in J_{i_1}(s)$ where $h_{j_2}(s) > 1$ — meaning $i_1$ is not the only winner in $q_{j_2}$. By Lemma B.5, if $w_{j_2}(s) < p$ then the residual of player $i_1$ is zero, in contradiction to equation 5. Therefore, we get that $w_{j_2}(s) \geq p > w_{j_1}(s)$.

Consider $i_t$, another winner in $q_{j_2}$. Suppose $i_t$ deviates by reallocating his resources from $q_{j_2}$ to $q_{j_1}$, aiming to win solely in the latter. Such a deviation would make $i_t$ the only winner in $q_{j_1}$, without affecting his rewards from other queries $q_j \in J_{i_t}(s) \setminus \{q_{j_2}\}$. Thus, this deviation would be profitable for $i_t$, in contradiction to the assumption that $s$ is an equilibrium. Therefore, we conclude that $i_1$ must be the unique winner in all his winning queries.

We define $I_u$ as the set of players who are unique winners in all the queries in which they are ranked highest. Under the Theorem's conditions, $I_u$ is non-empty. Let $\alpha = |I_u|$ represent the number of such winners, where $1 \leq \alpha \leq n$.

Let $k = \left\lfloor \frac{1}{p} \right\rfloor$ ($k \geq 2$). Under the false assumption that $w_{j_1}(s) < p$:

$$w_m(s) \leq w_{j_1}(s) < p$$

According to Lemma B.6, for each player $i$, $x_i(s)$ is bounded by their residual $y_i(s)$. Since $s$ is an equilibrium, Lemma B.4 states that for every player such that $q_m \notin J_i(s)$, the residual is bounded by $w_m(s)$. Combining, we get that for every player that is a loser in query $q_m$,

$$x_i(s) \leq y_i(s) \leq w_m(s) < p$$

If $h_m(s) \geq 2$, i.e. there are multiple winners in query $q_m$, then by Lemma B.5, $x_i(s) = y_i(s) = 0$ for each of the winners $i$ in query $q_m$. Thus, there exists a player $i_2 \in I_u$ for which the following holds:

$$\sum_{q_j \in J_{i_2}(s)} w_j(s_{-i_2}) = \sum_{q_j \in J_{i_2}(s):h_j(s)=1} w_j(s_{-i_2})$$

$$= \sum_{q_j \in J_{i_2}(s):h_j(s)=1} \max_{i \neq i_2} d_i^j$$

$$\leq \sum_{i \neq i_2} x_i(s)$$

$$= \sum_{i \in I_u \setminus \{i_2\}} x_i(s)$$

$$< \sum_{i \in I_u \setminus \{i_2\}} p$$

$$= (\alpha - 1) \cdot p$$

Consider the case where the number of players that win solely in all their queries is less than $k$, i.e. $\alpha \leq k$. We get that for player $i_2$,

$$\sum_{q_j \in J_{i_2}(s)} w_j(s_{-i_2}) + p < (\alpha - 1) \cdot p + p = \alpha \cdot p$$

$$\leq k \cdot p \leq 1$$

Rearranging, we obtain that the residual of player $i_2$ is larger that the peak value:

$$p < 1 - \sum_{q_j \in J_{i_2}(s)} w_j(s_{-i_2}) = y_{i_2}(s)$$

By Lemma B.4, player $i_2$ has a profitable deviation, in contradiction.

For the case where $\alpha \geq k + 1$, we show a similar result. However, there's a more nuanced point to consider. The residuals not allocated to winning are distributed across multiple queries involving multiple players. If these residuals are focused on a single player, he might not have a profitable deviation. However, this distribution implies that another player will be minimally affected by others in the queries they lose, and he will have an incentive to deviate.

Let $i_m$ be a winner in query $q_m$. We bound $x_{i_m}(s)$ using the upper bound from Lemma B.8,

$$x_{i_m}(s) \leq 1 - k \cdot w_m(s)$$

Recall that if $h_m(s) \geq 2$, $x_{i_m}(s) = 0$.

Now that we've bounded $x_i(s)$ for all players, we get the following:

$$\sum_{i \in I_u} \sum_{q_j \in J_i(s)} w_j(s_{-i}) \leq \sum_{i=1}^{n} \sum_{q_j \in J_i(s):h_j(s)=1} w_j(s_{-i})$$

$$\leq \sum_{i=1}^{n} x_i(s)$$

$$= x_{i_m}(s) + \sum_{i \in I_u \setminus \{i_m\}} x_i(s)$$

$$< 1 - k \cdot w_m(s) + (\alpha - 1) \cdot w_m(s)$$

$$= w_m(s) \cdot (\alpha - (k+1)) + 1$$

$$< p \cdot (\alpha - (k+1)) + 1$$

$$< p \cdot \alpha$$

Therefore, there exists a player $i_3 \in I_u$ such that:

$$\sum_{q_j \in J_{i_3}(s)} w_j(s_{-i_3}) < p$$

Rearranging, we see that by the definition of $y_{i_3}(s)$, $i_3$'s residual is:

$$y_{i_3}(s) = 1 - \sum_{q_j \in J_{i_3}(s)} w_j(s_{-i_3}) > 1 - p \geq p$$

By Lemma B.4, $i_1$ has a profitable deviation, by contradiction. □

We now show that if $p > \frac{1}{2}$ there is no pure equilibrium in the game when $n < m$.

LEMMA B.11. *Let $G = \langle n, m, p \rangle$ such that $n < m$ and $p > \frac{1}{2}$. Then $G$ doesn't have a pure equilibrium.*

PROOF. Following similar steps as in the proof of B.10, we can obtain that $w_m(s) \leq \frac{1}{2}$. If $w_m(s) > \frac{1}{2}$ then the number of queries in which each of the players win is exactly 1. However, this leads to $\sum_{i=1}^{n} |J_i(s)| = n < m$, which means there exists a query where all players publish $d_i^j = 0$. Every player would then have a profitable deviation, by contradiction.

Thus, every player can win in no more than two queries, i.e. $|J_i(s)| \leq 2$. Therefore $n < m \leq \sum_{i=1}^{n} |J_i(s)| \leq 2 \cdot n$, and the minimal winning value is at least $\frac{1}{2}$: $w_m(s) \geq \frac{1}{2}$. There exists a player $i_1$ such that $U_{i_1}(s) > 1$, therefore $w_m(s) = \frac{1}{2}$.

Overall, we get that $x_i(s) < \frac{1}{2}$ for every player $i$. So if $w_1(s) > \frac{1}{2}$, then the residual of the player $i_2$ that is the winner in query $q_1$ is:

$$y_{i_2}(s) = 1 - w_1(s_{-i_2}) > \frac{1}{2} = w_m(s)$$

By Lemma B.4, $i_2$ has a profitable deviation, by contradiction.

In conclusion, $w_1(s) = \ldots = w_m(s) = \frac{1}{2}$. Thus, for all players: $x_i(s) = 0$.

Going back to player $i_1$, recall that $U_{i_1}(s) > 1$. Hence, $i_1$ is the unique winner in at least one of the two queries in which he wins. Therefore we get that the residual of $i_1$:

$$y_{i_1}(s) > \frac{1}{2}$$

By Lemma B.4, $s$ is not an equilibrium - by contradiction. □

Following the specific case where a query had a sole winner with a winning value less than the peak value $p$, we now extend our analysis to cover all possible configurations of the strategy profile $s$. Specifically, we complete the proof for Lemma B.9:

LEMMA B.9. *Let $G = \langle n, m, p \rangle$ such that $n < m$ and $s$ be a pure equilibrium profile. Then for every query $q_j \in Q$,*

$$w_j(s) \geq p$$

PROOF. Let $k = \lfloor \frac{1}{p} \rfloor$, and assume by contradiction that $w_m(s) < p$.

According to Lemma B.11, if $p > \frac{1}{2}$ then the game doesn't possess a pure equilibrium. Given that $s$ is a pure equilibrium profile, it necessarily follows that $p \leq \frac{1}{2}$, thereby ensuring that $k \geq 2$. If there exists a query $q_j'$ such that $w_j'(s) < p$ and $h_j'(s) = 1$, then

by Lemma B.10 $w_m(s) \geq p$. Hence, for all queries $q'_j$ such that $w'_j(s) < p$ it holds that $h'_j(s) \geq 2$.

If there exists any query $q'_j$ such that $w'_j(s) < p$ and $h'_j(s) = 1$, then by Lemma B.10, we should have $w_m(s) \geq p$. Therefore, any query $q'_j$ where $w'_j(s) < p$ must have $h'_j(s) \geq 2$.

We now argue that the number of winning queries $|J_i(s)|$ for any player $i$ is at most $k$. Assume by contradiction that there exists a player $i_1$ that has $|J_{i_1}(s)| \geq k + 1$.

Define $T$ as the set of queries where $i_1$ wins but the winning value is less than $p$:

$$T := \{q_j \in J_{i_1}(s) : w_j(s) < p\}$$

Under the false assumption that $|T| > 1$, let $q_{j_2} \in J_{i_1}(s) \setminus T$.

By Lemma B.5, the residual $y_{i_1}(s) = 0$, precluding $i_1$ from being the unique winner in any query. Therefore, $h_{j_2}(s) \geq 2$, and $i_1$ can profitably deviate by shifting his efforts from query $q_{j_2}$ in order to win solely in all queries in $T$. If $|T| \geq 2$, this deviation becomes profitable, leading to a contradiction. Thus, we conclude $|T| \leq 1$.

By the definition of $T$, for every query $q_j \in J_{i_1}(s) \setminus T$: $d_{i_1}^j = p$. Therefore the size of $J_{i_1}(s) \setminus T$ is upper bounded by $k$. Overall, $|J_{i_1}(s)| = k + 1$.

By Lemma B.5, we know for player $i_1$ it holds:

$$0 = y_{i_1}(s)$$
$$= 1 - \sum_{q_j \in J_{i_1}(s)} w_j(s_{-i_1})$$
$$= 1 - \sum_{q_j \in J_{i_1}(s)} w_j(s)$$
$$= 1 - w_m(s) - k \cdot p$$

Rearranging gives that the minimal winning value over all queries is:

$$w_m(s) = 1 - k \cdot p$$

Let $i_2$ be a player that is not a winner in query $q_m$, so $q_m \notin J_{i_2}(s)$. Using similar arguments as for $i_1$, there exists a query $q_{j_2} \in J_{i_2}(s)$ where $w_{j_2}(s) = p$. If $h_{j_2}(s) \geq 2$, then $i_2$ has a profitable deviation to a strategy where he wins solely in query $q_m$ instead of sharing his reward for query $q_{j_2}$. Hence, $i_2$ is a unique winner in $q_{j_2}$, and therefore $w_j(s_{-i_2}) < w_m(s)$. This applies to every for every query $q_{j_2} \in J_{i_2}(s)$ where $w_{j_2}(s) = p$, and $i_2$ is the unique winner in all queries where he is the winner.

Therefore, the residual of $i_2$ is:

$$\sum_{q_j \in J_{i_2}(s)} w_j(s_{-i}) < \min\{(n-2), k\} \cdot (1 - k \cdot p) < k \cdot (1 - k \cdot p) \leq 1 - p$$

And then the residual of player $i_2$ is:

$$y_{i_2}(s) = 1 - \sum_{q_j \in J_{i_2}(s)} w_j(s_{-i}) > p$$

By Lemma B.4, $i_2$ has a profitable deviation - by contradiction.

So $|J_i(s)| = k$ for every player $i$. Now, every player $i_m$ that is a winner in query $q_m$ has a profitable deviation, since $y_{i_m}(s) > 0$ (Lemma B.5). Therefore $w_m(s) \geq p$. □

We showed that for $m > n$, the value of the winners in all queries is at least $p$. Next, we show that there are at most $n$ queries in which there is a unique winner. Specifically, each player can be the unique winner no more than one query.

LEMMA B.12. *Let $G = \langle n, m, p \rangle$ such that $n < m$, and let $s$ be a pure equilibrium profile. Then every player is the unique winner in at most one query.*

PROOF. Let $k = \left\lfloor \frac{1}{p} \right\rfloor$. Lemma B.9 states that for every query $q_j$, $w_j(s) \geq p$. In conjunction with Lemma B.7, this allows us to conclude that $|J_i(s)| = k$, meaning each player wins exactly $k$ queries.

Assume by contradiction that there exists some player $i_1$ that is the unique winner in two queries $q_{j_1}, q_{j_2}$. Let $q_{j_3} \in Q$ be a query in which $i_1$ is a loser, i.e. $q_{j_3} \notin J_{i_1}(s)$.

Since $p > \frac{1}{k+1}$, there exists some $\epsilon > 0$ s.t. the following deviation is well defined:

$$d_t^j = \begin{cases} (1 - p \cdot k) + \epsilon & \text{if } q_j \in \{q_{j_1}, q_{j_2}\} \\ p & \text{if } q_j \in J_i(s) \cup \{q_{j_3}\} \setminus \{q_{j_1}, q_{j_2}\} \\ 0 & \text{otherwise} \end{cases}$$

After deviating to $d_t$, player $i_1$'s position remains unchanged in all queries $q_j \in J_{i_1}(s)$. Additionally, $i_1$ now becomes at least one of the winners in $q_{j_3}$. Consequently, his utility increases by at least $\frac{1}{n}$, which signifies a profitable deviation. This contradicts the initial state $s$ being a pure equilibrium. □

Building on the necessary conditions established in Lemmas B.9 and B.12, we demonstrate that the game lacks pure equilibrium when $p > \frac{1}{\lceil \frac{2 \cdot m}{n} - 1 \rceil}$. Then, we demonstrate that when $p \leq \frac{1}{\lceil \frac{2 \cdot m}{n} - 1 \rceil}$, there exists an equilibrium in the game, and it can be achieved by Algorithm 1.

THEOREM 3.4. *The game $G = \langle n, m, p \rangle$ with $n < m$ has a pure Nash equilibrium iff $p \leq \frac{1}{\lceil \frac{2 \cdot m}{n} - 1 \rceil}$.*

PROOF. Let $k \in \mathbb{N}$ such that $\frac{1}{k+1} < p \leq \frac{1}{k}$. We need to show that $G$ possess a pure equilibrium iff $k \geq \lceil \frac{2 \cdot m}{n} \rceil - 1$.

Lemma B.11 establishes that no equilibrium exists for $p > \frac{1}{2}$, so we consider scenarios where $k \geq 2$. From Lemma B.9, it follows that for every query $q_j$, the winning value $w_j(s) \geq p$. This determines that each player $i$ is a winner in exactly $k$ queries: $|J_i(s)| = k$.

Let $t$ be the number of queries in which $h_j(s) = 1$:

$$t = \sum_{q_j \in Q} \mathbb{1}_{h_j(s)=1} = \sum_{q_j \in Q: h_j(s)=1} h_j(s)$$

Considering the total number of winning positions, we have:

$$n \cdot k = \sum_{j=1}^{m} h_j(s)$$

$$= \sum_{j:h_j(s)=1}^{m} h_j(s) + \sum_{j:h_j(s)\geq 2}^{m} h_j(s)$$

$$= t + \sum_{j:h_j(s)\geq 2}^{m} h_j(s)$$

$$\geq t + (m-t) \cdot \min_{j:h_j(s)\geq 2} h_j(s)$$

$$= t + 2 \cdot (m-t)$$

$$= 2 \cdot m - t$$

Lemma B.12 restricts each player to being the unique winner in at most one query, implying $t \leq n$. Rearranging, we get that the value of $k$ is:

$$k \geq \frac{2 \cdot m - t}{n} \geq \frac{2 \cdot m}{n} - 1$$

Since $k \in \mathbb{N}$, the necessary conditions for equilibrium require that $k \geq \lceil \frac{2 \cdot m}{n} \rceil - 1$. Consequently, if $p > \frac{1}{\lceil \frac{2 \cdot m}{n} - 1 \rceil}$ then a pure Nash equilibrium cannot exist in the game.

Next, we show that if $k \geq \lceil \frac{2 \cdot m}{n} \rceil - 1$, the game has an equilibrium, which can be achieved by Algorithm 1. Let $s_{\text{alg}}$ denote the strategy profile obtained by Algorithm 1.

At the conclusion of the algorithm, each player wins in exactly $k$ queries. Since $k \geq \lceil \frac{2 \cdot m}{n} - 1 \rceil$, it ensures that at least one player selects each query $q_j$, resulting in $w_j(s) = p$ for every query. Throughout the algorithm, players choose the query with the fewest current winners, maintaining the condition:

$$\max_{q_j \in Q} h_j(s) - \min_{q_j \in Q} h_j(s) \leq 1$$

By the end of the algorithm, the number of winners in each query $q_j$ is bounded by:

$$\left\lfloor \frac{n \cdot k}{m} \right\rfloor \leq h_j(s_{\text{alg}}) \leq \left\lceil \frac{n \cdot k}{m} \right\rceil$$

Assuming a lexical tie-breaking rule, no player ends up being the sole winner in more than one query. If any player $i_t$ were to deviate with a strategy $d_t$, they could not potentially win in more than $k$ queries due to $w_j(s) \geq p > \frac{1}{k+1}$. Introducing this strategy would only increase the number of winners in any query in which $i_t$ didn't previously won by one. Consequently, such a change would not improve the player's utility, as it does not offer a better payoff than the current strategy. Thus, no deviation from $s_{\text{alg}}$ yields a higher utility, and $s_{\text{alg}}$ is a pure Nash equilibrium. □