# OpenReview forum: "Ranking-Incentivized Document Manipulations for Multiple Queries"
_ACM.org/SIGIR/ICTIR/2024/Conference — ICTIR 2024_

### Official Review · Reviewer_1bcy · 2024-05-15

**Rating:** 2
**Confidence:** 4

**Objective Part Of Review:**

## Soundness:

The paper is robust in terms of its problem statement, methods description, results, and substantiated claims. It demonstrates a thorough approach to the research question and methodology, and I found no significant issues in these aspects.


## Presentation:

As someone not specializing in this field, I found the paper, especially the theoretical aspects in section 3, quite dense yet comprehensible. The authors have managed to present complex ideas and proofs in a way that, despite requiring some effort and back-and-forth reading, remains accessible. This indicates a high level of clarity in the paper's structure and delivery.


## Difficulty:

The "method" seems to be dual-layered: it involves both a theoretical analysis of the multi-query competitive retrieval problem using game theory principles and practical experiments involving ranking competitions with students acting as simulated publishers. Each component appears challenging and well-executed, making it difficult to determine which aspect was more demanding. The methodology is robust, with no apparent shortcomings in its execution.


## Impact:

Regarding the paper’s impact, I am somewhat tentative, given my limited expertise in this specific area. However, considering the soundness and clarity of the presentation, it seems that the paper could contribute significantly to its field. It would be beneficial to provide more insight into how this work advances the current understanding or technology.


## Related Work:

The review of related literature seems comprehensive, with all pertinent studies included. The transition from single-query to multiple-query contexts is a crucial evolution in the field and is well-addressed. However, reference [25], which seems pivotal, could benefit from a deeper discussion to highlight its relevance and the advances made from it.


## Scope regarding Topic and Methodology:

The paper appropriately covers the intended scope concerning both the topic and the methodology.

**Subjective Part Of Review:**

I'd like to suggest a few points for the authors to consider:

- Incorporating examples and figures could enhance comprehension for readers outside this field, although the current presentation is adequate.
- There seems to be a significant disconnect between the theories presented in Section 3 and the subsequent experiments. While each component is valuable on its own, their integration does not clearly support the paper’s conclusions. It would be beneficial to illustrate how these elements interact more effectively.
- The paper mentions using crowd workers to evaluate the relevance of documents written by students, yet I did not observe these evaluations being utilized in the experiments. Could you clarify if I've overlooked something?
- Most claims and conclusions in the paper lack explicit implications for the field, which is surprising given the relevance of this work to modern search engines. It might be fruitful to discuss potential implications for both publishers and search engine providers, possibly suggesting ways they could intervene.

---

### Official Review · Reviewer_BzFc · 2024-05-16

**Rating:** 2
**Confidence:** 3

**Objective Part Of Review:**

The authors contribute a theoretical and empirical study of document manipulations intended to improve ranking for multiple queries. They study document modifications in a game theoretical setting by organizing ranking competitions. This article is an extension on previous work that only considers ranking optimization for a single query. Some assumptions are taken from the literature (e.g. disclosure of the retrieval function, and using a single-peaked ranking function).

The authors show that while there is a Nash equilibrium in the one-query ranking game setting, it is not necessarily there in the multi-query setting. A characterization of when this is/is not the case is also provided.  Any claims made by the authors seem to be rooted in empirical data, and the target groups of students were large enough to draw conclusions from.

The paper is clear and concisely written, and an interesting case study that is a valuable addition to the conference in my view.

Small remarks: “RBO” is never defined before use. Typo in last paragraph 6.2 “anzlye” → “analyze”.

**Subjective Part Of Review:**

I think generally the strength of this paper is its extensive comparisons and experiments. Many different metrics, settings and configuration were tried and reported on. With the rationale that parties doing document optimization (e.g. SEO companies) are also using AI tools nowadays, these were also included in the study. The difference in document modifications with vs without AI tools was an interesting extra finding. While the number of abbreviations in the paper is a bit staggering, everything is clearly defined and referred to.

---

### Official Review · Reviewer_jPYA · 2024-05-23

**Rating:** 1
**Confidence:** 3

**Objective Part Of Review:**

The paper discusses a novel and relevant problem in the field of information retrieval, providing theoretical and empirical analysis of ranking-incentivized document manipulations for multiple queries. It is well-structured, with clear problem statements, theoretical analysis, experiment settings, and results discussion. The claims are supported by evidence. The limitation is that the analysis features are somehow traditional and may be far away from the real search engine scenarios.

**Pros:**

1.	The paper is written fluently and clearly claims the problems and analysis.

2.	The theoretical analysis is solid and rigorous.

3.	The empirical study is sufficient and the introduction of generative AI tools to perform document modification is interesting and brings some insights for analyzing the role and impact of generative AI in future information retrieval.

**Cons:**

1.	The ranking functions and features selected to analyze are traditional and rule-based, which may be far from real web search scenarios when users modify documents to fit the online search engine. It will be better to introduce generative AI models to provide more flexible and advanced evaluation features.

2.	Whether the setting of different ranking functions is unfair? This paper analyzes users’ editing behaviors for multiple queries in two different ranking function settings, i.e. feature-based LTR method and neural ranking algorithm, BERT. However, the BERT-similarity is also used as one of the features of the LTR method. It seems like the basic performance of these two ranking functions is different and the LTR method is inevitably more robust than the BERT-based model.

**Subjective Part Of Review:**

Overall, the paper is well-structured, and the discussed topic is relevant to the ICTIR community. The method is original and the corresponding experimental results are sufficient and bring some interesting insights.

---

### Official Review · Reviewer_13EE · 2024-05-23

**Rating:** -1
**Confidence:** 4

**Objective Part Of Review:**

The paper carries the work forward from the papers cited [12, 32], and establishes that Nash Equilibrium exists for multiple queries, as well. The paper is, in general, well written - a bit heavy on the theory, most of which is moved to the appendix (which is a good thing).

The experiments are conducted on the standard UQV dataset, which is also a plus.

A few concerns about the lack of clarity of using LLMs:
1. I didn’t quite get the idea of the rationale of using an LLM (GPT) for generating a relevant document.
2. Why wasn’t a known relevant document used for a backstory that was created from a TREC Robust query?
3. Why is it that an LLM generated relevant document is hypothesized to be more diverse?

**Subjective Part Of Review:**

What I like to see in the paper is an explicit list of research questions that clearly sets the expectations straight from the experiments and their results. What is it that we want to verify empirically and how much of that relates back to the theory should be clearly described.

Moreover, while I understand that the paper is different from [12, 32] because it models multiple queries, there should be a more clear comparison on the analysis proposed in [12, 32] vs. the ones proposed in this paper. For example, does the current paper, in some sense, generalize the theoretical analysis framework of [12]? If so how? What are the new variables and the new interactions between them? What are the new assumptions?


About the experiments, with human subjects - I see a potential ethics red flag because the following, IMO, doesn’t appear to be fair to the students.

“To encourage students to participate and modify their documents so as to attain high rankings in the competitions, we offered bonus points for their course grades."  --- this, IMO, is putting stress on the students and where is the guarantee that such a stressful environment will actually result in a productive outcome?

---

### Meta-Review · Area_Chair_pvFg · 2024-05-29

**Recommendation:** Accept (Oral)
**Confidence:** 4

**Metareview:**

This paper proposes an extension of a previous approach proposed by the authors, which has the objective of incentivizing the modification of documents by their authors by means of improved rankings. The formal ground is game theory. The proposed extesion applies to considering multiple queries instead of single query settings.
The paper reports interesting researcg, which can raise discussions in tha audience. The  presentation needs some clarifications to improve the final version of the manuscript: 1) better  explain the link between the model related formulation and the experimental part; 2) some aspects related to the experimental evaluations need clarification (e.g. crowdsourcing); 3) in general the authors are recommended to account for the reviewers’ comments in preparing the final version of the paper.